

# Vortex Preconditioning of the 2021 Sudden Stratospheric Warming:
# Barotropic/Baroclinic Instability Associated with the Double Westerly
# Jets
Ji-Hee Yoo[1], Hye-Yeong Chun[1], Min-Jee Kang[2]
[1]Department of Atmospheric Sciences, Yonsei University, Seoul, 03722, South Korea
[2]School of Earth and Environmental Sciences, Seoul National University, Seoul, 08826, South Korea
*Correspondence to*: Hye-Yeong Chun (chunhy@yonsei.ac.kr)
**Abstract.** This study explores the abrupt split of the polar vortex in the upper stratosphere prior to a recent sudden
stratospheric warming event on 5 January 2021 (SSW21) and the mechanisms of vortex preconditioning by using the
Modern-Era Retrospective Analysis for Research and Applications version 2 (MERRA2) global reanalysis data. SSW21
is preceded by the highly distorted polar vortex that was initially displaced off the pole but eventually split at the onset
date. Vortex splitting is most significant in the mid-stratosphere (1 hPa altitude) accompanied by the anomalous growth
of westward-propagating planetary waves (PWs) of zonal wavenumber (ZWN) 2 (WPW2). While previous studies have
suggested the East Asian trough as a potential source for the abnormal WPW2 growth, the prominent westward-
propagating nature cannot be explained satisfactorily by the upward propagation of the quasi-stationary ZWN2 fluxes
in the troposphere. More importantly, WPW2 exhibits an obvious in-situ excitation signature within the barotropically
and baroclinically destabilized stratosphere, dominated by the easterlies descending from the stratopause containing the
WPW2 critical levels. This suggests that the vortex split is attributed to the WPW2 generated in situ within the
stratosphere via instability. Vortex destabilization is achieved as the double-jet structure consisting of a subtropical
mesospheric core and a polar stratospheric core develops into SSW21 by encouraging the anomalous dissipation of the
upward-propagating tropospheric ZWN1 PWs. This double-jet configuration is likely a favorable precursor for SSW
onset, not only for the SSW21 but generally for most SSWs, through promoting the anomalous growth of unstable PWs
as well as the enhancement of the tropospheric PW dissipation.

## 1 Introduction
Sudden stratospheric warming (SSW) is a dramatic stratospheric phenomenon where the cold and strong westerly polar
night jet (PNJ) rapidly decelerates or even reverses to easterly with an enormous warming within a week (Matsuno,
1971). During SSW, the polar vortex is largely displaced away from the pole and/or split into two vortices (Charlton
and Polvani, 2007, CP07). The impact of SSW is not limited to the polar stratosphere but extended into the mesosphere
and above, causing significant changes in the residual circulations (Limpasuvan et al., 2016; Siskind et al., 2010), the
distributions of chemical constituents such as ozone (Manney et al., 2009; Pedatella et al., 2018), and the atmospheric
tides both in the Northern and Southern hemispheres. The dramatic temperature and wind perturbations during SSWs
also descend into the troposphere, thereby altering the storm tracks which are closely tied to the surface weather patterns
(Baldwin and Dunkerton, 2001; Hitchcock and Simpson, 2016).
SSW has been recognized as a manifestation of the interaction between the vertically propagating planetary waves (PWs)
and stratospheric mean-flow. This is primarily driven by the upward-propagating anomalous tropospheric wave pulses,
which can provide sufficient wave forcings to breakdown the polar vortex (Matsuno, 1971), and/or preconditioning of
the stratosphere that focuses the tropospheric wave fluxes—not need to be anomalously strong—into the polar
stratosphere (Birner and Albers, 2017; Palmer 1981). The preconditioning perspective has also been discussed in terms
of the spontaneous wave explosion within the stratosphere (Plumb, 1981) as the polar vortex tunes itself toward the
explosive wave-growth point, such as resonance (Albers and Birner, 2014, AB14) or barotropic/baroclinic (BT/BC)
instability (Sato and Nomoto, 2015). Recent supports for the vortex preconditioning have been identified from
observational (AB14; Iida et al., 2014) and modeling (Rhodes et al., 2021, RLO21) studies on the split-type SSW of
January 2009 (SSW09). Such self-tuned SSWs are characterized by nearly instantaneous wave amplification throughout
the entire stratosphere at the SSW onset. Within this context, AB14 interpreted the explosive growth of stratospheric
wave activities as a manifestation of vortex breakdown, not the cause of SSW.
The latest major SSW took place on 5 January 2021 (SSW21), exhibiting the highly distorted polar vortex that was
initially displaced off the pole but eventually split at the onset date. During the prewarming period, an initial zonal



wavenumber (ZWN) 1 pulse followed by a ZWN2 pulse was identified in the tropopause, suggesting their contributions
to the observed vortex collapse (Cho et al., 2022; Lu et al., 2021; Rao et al., 2021). Lu et al. (2021) and Rao et al. (2021)
related the intensification of the Aleutian low and the North Atlantic high in late December 2020 to the enhanced
tropospheric ZWN1 flux and that of the East Asian trough developed in early January 2021 to the succeeding ZWN2
flux. By performing numerical experiments, Cho et al. (2022) showed that the tropospheric ZWN1 pulse is attributed
primarily to the North Pacific bomb cyclones that deepened the Aleutian low with a minor contribution from the Ural
blocking.
This study expands upon previous research on SSW21 by examining the prewarming evolution of the vortex throughout
the entire stratosphere, rather than solely in the region below 10 hPa conducted by most of previous studies on SSW21.
We found that the most significant vortex split occurs in the mid-stratosphere (1 hPa). However, the anomalous
stratospheric ZWN2 PWs (PW2) amplification responsible for this split cannot be explained by the concomitantly
enhanced tropospheric ZWN2 fluxes. Therefore, this study explores vortex preconditioning in the context of the
spontaneous PW2 explosion while addressing two questions: i) What is the source of the stratospheric PW2
amplification? ii) How does the stratospheric vortex evolve toward the wave-growth point? To our knowledge, this is
the first study to explore the role of vortex preconditioning in SSW21, providing more comprehensive accounts of the
dynamics leading to SSW21.

## 2 Data and Analysis Methods
### 2.1 The MERRA2 reanalysis data
We use the Modern-Era Retrospective analysis for Research and Applications, version 2 (MERRA2) reanalysis data
with a horizontal resolution of 0.625° ×0.5° (longitude × latitude) and a temporal resolution of 3 hours from the surface
to an altitude of 0.1 hPa (Gelaro et al., 2017) covering 42 years (1980–2021). All results in this study are based on the
daily average.

### 2.2 Analysis methods
The Eliassen-Palm flux (EP-flux) and their divergence (EPFD), representing the wave activity flux and wave forcing,
respectively, are calculated based on the following formulation (Andrews et al., 1987):

$$\boldsymbol{F} = \left(F^{\phi}, F^{z}\right) = \rho_0 a \cos\phi \left(-\overline{u'v'} + \bar{u}_z \frac{\overline{v'\theta'}}{\bar{\theta}_z}, \left[f - \frac{1}{a\cos\phi}(\bar{u}\cos\phi)_\phi\right]\frac{\overline{v'\theta'}}{\bar{\theta}_z} - \overline{u'w'}\right), \tag{1}$$

$$\nabla \cdot \boldsymbol{F} = \frac{1}{a\cos\phi}\frac{\partial}{\partial\phi}\left(F^{\phi}\cos\phi\right) + \frac{\partial F^{z}}{\partial z}, \tag{2}$$


where $\phi$ and $z$ are the latitude and log-pressure height, respectively, $\rho_0$ is the reference density, $a$ is the mean Earth's
radius, and $f$ is the Coriolis parameter. $u$, $v$, and $w$ are the zonal, meridional, and vertical wind components, respectively,
and $\theta$ is the potential temperature. The overbar and prime represent the zonal-mean and the departure from the zonal-
mean, respectively. $\boldsymbol{F}$ is the EP-flux vector, where $F^{\phi}$ and $F^{z}$ are the meridional and vertical components, respectively.
EPFD corresponds to $(1/\rho_0 a\cos\phi)\nabla\cdot\boldsymbol{F}$.
BT/BC instability is evaluated by using the meridional gradient of the quasi-geostrophic potential vorticity (QGPV,
Andrews et al., 1987):

$$\bar{q}_y = \beta - \bar{u}_{yy} - \frac{1}{\rho_0}\left(\rho_0\frac{f^2}{N^2}\bar{u}_z\right)_z, \tag{3}$$






where $\bar{q}$, $\beta$, and $N$ denote the zonal-mean QGPV, the meridional derivative of $f$, and the Brunt–Väisälä frequency,
respectively. The necessary condition for BT/BC instability is that the generally positive $\bar{q}_y$ associated with the
wintertime circulation becomes negative (Salby, 1996). In Section 3, we refer to the sum of the first two terms on the
right-hand side as the "barotropic term", while the third term as "baroclinic term".

**3 Results**
**3.1 Wind and temperature changes during SSW21**
Figure 1a shows the time-evolutions of the zonal-mean zonal wind at 60°N and polar-cap temperature over 60–90°N
during the development of SSW21. Remarkably, a reversal of the zonal-mean westerlies appears first in the lower
mesosphere on 1 January and descends to 10 hPa within 4 days, leading to the onset of major SSW21 (CP07). It is
preceded by the enormous deceleration of PNJ by ~108 m/s and a rapid 20 K warming in the mid-stratosphere (~1 hPa)
within 8 days (28 December–4 January). Such a decrease (increase) in the zonal wind (temperature) is statistically
significant at the 99% confidence level. Anomalous easterlies and warming descend into the troposphere and persist for
longer than 20 days, which is much longer than the average persistence (~8 days) following SSWs in the reanalysis and
CMIP models (Rao and Garfinkel, 2021).

**3.2 Anomalous Enhancement of the Stratospheric PW2**
SSW21 is manifested by the polar vortex being severely displaced from the pole and ultimately split into two just before
the onset. Associated PW activities are revealed in Figure 1b, which describes the time-evolutions of the geopotential
height (GPH) amplitudes of PW1 and 2 at 60°N. As Lag = -1 is approached, the predominant PW1 amplitude drastically
decreases, while the PW2 amplitude appreciably increases having the statistically significant positive anomaly at the
95% confidence level at 1–3 hPa. From Lag = -2 to Lag = 1, PW2 dominates in the mid-to-upper stratosphere above 3
hPa. Given the prevalent dominance of PW1 in the high-latitude winter stratosphere (Andrews et al., 1987; Matsuno
1970), predominant PW2 activity observed in this case and other split-type SSWs is a notable feature. Evidenced in
Figure 1c, which compares the polar-stereography series of the horizontal wind speed and the GPH anomaly at 1 and
10 hPa, the vortex split is more pronounced in the mid-stratosphere than in the lower stratosphere, where PW1 have
surpassed PW2 (Figure 1b).
Previous studies have suggested that the vortex split is attributed to the enhanced tropospheric ZWN2 fluxes entering
the stratosphere, as evidenced by peak pulses of the ZWN2 eddy heat flux averaged over 45–75°N at 100 hPa during 1–
5 January. However, this period nearly coincides with that of remarkable PW2 amplification in the mid-stratosphere
(Figure 1b). This implies that the increased tropospheric fluxes must have instantaneously propagated up to ~28 km
within the mid-to-upper stratosphere, which is highly questionable. Therefore, we examine whether the large
tropospheric pulses are traceable to the upper stratosphere at the standard group velocity for vertically propagating PW2.
Figure 2a illustrates the time-height cross section of the vertical component of EP-flux (EPFz) of PW2 in 45–75°N and
the three identical vectors with a slope of 5.5 km/day, which corresponds to the theoretical group velocity of the
vertically propagating Rossby waves of ZWN2 (Esler and Scott, 2005). For comparison purpose with previous studies,
the time-series of eddy heat flux ($\overline{v'T'}$) of ZWN1 and 2 in 45–75°N at 100 hPa are also presented below.
While $\overline{v'T'}$ of ZWN1 reduces, that of ZWN2 increases from 28 December (Lag = -8), attaining a magnitude 1 STD
greater than the climatology (but not significant) during 1–5 January. The theoretical prediction of Rossby waves'
vertical propagation well matches the vertical propagation of EPFz below 5 hPa, indicating that the bulk of ZWN2
fluxes propagate upward (AB14). However, as evidenced by the third vector, these waves could approach the upper
stratosphere ~2 days after the onset date via upward propagation. This implies that the statistically significant PW2
amplification in the upper stratosphere in Lag = -3–Lag = -1 (Figure 1b) cannot originate from the anomalous injection
of the tropospheric wave activity during the same period.
More importantly, EPFz is not continuous above 5 hPa and exhibits apparent divergences with the downward EPFz
(negative) below the region of upward EPFz (positive) around 3 hPa from Lag = -5 to Lag = -3. This suggests a potential
for the in situ PW2 generation within the stratosphere. Despite the disappearance of downward EPFz after Lag = -2, the





divergence continues with the locally maximized upward EPFz in 10–3 hPa. In this view, subsequent statistically
significant enhancement in the upward EPFz (exceeding 99% confidence level) above the divergence altitude could be
a consequence of the upward propagation of the in situ generated PW2.
The evolution of the PW2 GPH in 45–75°N, as a function of zonal phase speed and time at the three altitudes depicted
in Figure 2b, supports this perspective. During the strengthening period of ZWN2 $\overline{v'T'}$ (Lag = -8–0), the tropospheric
PW2 (100 hPa) has a quasi-stationary nature, whereas the stratospheric PW2 (1–3 hPa) has prominent westward phase
speeds of 10–30 m/s (WPW2). The stratospheric WPW2 cannot be explained solely by the upward propagation of the
quasi-stationary tropospheric PWs.

**3.3 In situ Source of the Stratospheric WPW2: BT/BC Instability**
To examine the potential source of the stratospheric PW2, we first investigate EP-fluxes and EPFD of PW2 during the
WPW2 amplification period (1–5 January, Figure 3a). In this analysis, the overall PW2 behavior is investigated, not
exclusively for WPW2.
Throughout the period, significantly anomalous divergence of EP-fluxes (positive EPFD) appears, developing with the
rapidly intensifying easterlies. This demonstrates the spontaneous PW2 emanation within the stratosphere, which is
associated with the background flow: positive EPFD first appears between the easterlies extending from the equatorial
stratosphere and the polar jet core (Lag = -4). As the polar stratosphere becomes dominated by the descending
stratopause easterlies, the divergence is also enlarged towards 10 hPa and simultaneously intensified, exceeding 50
m/s/day at Lag = -2. While the easterlies further strengthen after that, the divergence area narrows below the jet core.
Nevertheless, the PW2 fluxes evolving along their propagation have magnitudes comparable to or even greater than the
previous ones. The upward propagating tropospheric fluxes, on the other hand, converge before reaching the easterlies,
imposing westward forcing. This is consistent with their quasi-stationary nature, which is inhibited by the zero-wind
line.
As a plausible in situ source for the stratospheric PW2, BT/BC instability is examined. Figures 3b–3d present the
latitude-height cross sections of $\bar{q}_y$ and the barotropic and baroclinic terms of Equation (3), respectively. Negative $\bar{q}_y$
satisfying the BT/BC instability condition emerges around the positive EPFD areas during the overall period. Similar to
the positive EPFD, this instability is exacerbated by the developing easterlies, attributed to both the barotropic and
baroclinic terms. The strengthening easterlies induce the positive $\bar{u}_{yy}$ along their maxima, which dominates the positive
$\beta$, leading to the vertically oriented negative barotropic term (Figure 3c). Concurrently, the baroclinic term becomes
negative from below the easterly core (Figure 3d). To elucidate the dominant factors that make the baroclinic term
negative, the third term of the right-hand side of Equation (3) is expanded as follows:

$$-\frac{1}{\rho_0}\left(\rho_0\frac{f^2}{N^2}\bar{u}_z\right)_z = f^2\left[\frac{1}{H}\frac{1}{N^2}\bar{u}_z + \frac{1}{N^4}\frac{dN^2}{dz}\bar{u}_z - \frac{1}{N^2}\bar{u}_{zz}\right],$$
(4)


where $H$ is the scale height (7 km).
Figures 4a–4c present the latitude-height cross sections of the first, second, and third terms of the right-hand side of
Equation (4), respectively, divided by $f^2$ during the vortex destabilization period (1–5 January). It shows that the
negative baroclinic term is attributed to both the first and third terms within the developing easterlies in the polar
stratosphere, with an insignificant compensation by positive value from the second term.
Figures 4d–4g show the latitude-height cross sections of the inverse of the squared Brunt–Väisälä frequency $1/N^2$, the
vertical gradient of the zonal-mean zonal wind $\bar{u}_z$, the vertical gradient of the Brunt–Väisälä frequency $dN^2/dz$, and
the vertical curvature of the zonal-mean zonal wind $\bar{u}_{zz}$, respectively, those consist of the three terms on Equation (4).
The negative first term is induced by the negative $\bar{u}_z$ (Figure 4e) as the subtropical stratospheric easterlies propagate to
the polar stratopause (the polar stratopause easterlies descend into the lower stratosphere) on 1 January (2–5 January).
This negative $\bar{u}_z$ along with the negative $dN^2/dz$ (Figure 4f) makes the second term positive below the easterly jet core.
The negative third term, which is maximized above the easterly jet core, is caused by the strong positive $\bar{u}_{zz}$ (Figure 4g)



under relatively small $1/N^2$ (Figure 4e). Therefore, we conclude that the negative baroclinic term is attributed to the
negative $\bar{u}_z$ (positive $\bar{u}_{zz}$) below (centered at) the easterly jet core. Above findings suggest that the developing easterlies
cause WPW2 excitation by encouraging strong shear instabilities. These findings align with the numerical study by
Dickinson (1973): To serve instability as a source for PWs of a certain zonal phase speed $C_x$, the region must include a
critical layer where the zonal-mean zonal wind matches $C_x$. The presence of WPW2 critical levels near the in situ PW2
generation region is confirmed by the range of easterlies (-40–0 m/s) encompassing that of PW2's $C_x$ in the mid-
stratosphere (1–3 hPa, Figure 2b). The collocation of negative $\bar{q}_y$, the emergent PW2, and their critical levels
demonstrates that WPW2 grows by extracting energy from the unstable flow.
Yamazaki et al. (2021) found similar bursts of quasi-4-day WPW2s originating from the unstable stratosphere beyond
their critical level during the major SSWs in 2009, 2013, 2018, and 2019. Regarding the appearance of eastward-
propagating PWs of ZWN2 (EPW2) in the mesosphere before the SSW09 onset, Iida et al. (2014) also suspected in situ
generation via BT/BC instability in the westerly flow regime. RLO21 confirmed this possibility by identifying the
existence of the EPW2 critical level, but they interpreted EPW2 emergence as the over-reflection of the tropospheric
PW2 propagating upward. We explore the possibility of over-reflection for the amplified WPW2 by examining the
squared refractive index ($n^2$):

$$n^2 = \left[\frac{\bar{q}_\phi}{a(\bar{u} - C_x)} - \left(\frac{k}{a\cos\phi}\right)^2 - \left(\frac{f}{2NH}\right)^2\right]a^2. \tag{5}$$


Here, we set the zonal wavenumber $k = 2$ and the zonal phase speed $C_x = -10$ m/s, which corresponds to the identified
WPW2 peak in Figure 2b.
Figure 5 presents the latitude-height cross sections of the regions of negative $\bar{q}_y$ and positive $n^2$ with PW2 EP-fluxes
and EPFD in 1–5 January 2021. On 2 January, the over-reflection signal that bears a resemblance to the illustration in
Figure 1 in RLO21 is identified. Following the waveguide (orange hatched), the upward-propagating WPW2 are allowed
to reach the unstable region (mint shaded) where the critical level of WPW2 ($C_x = -10$ m/s) is located. Leaving behind
a strong EP-flux divergence region, downward PW2 EP-flux vectors point away from the evanescent region of negative
$n^2$ (without orange hatched), which is formed by the negative $\bar{q}_y$ and positive $\bar{u} - C_x$. These downward vectors can be
interpreted as the over-reflection of upward-propagating WPW2. This is consistent with the local downward EPFz below
the upward EPFz in Figure 2a. The positive $n^2$ region associated with the transition from positive to negative $\bar{u} - C_x$
under the negative $\bar{q}_y$ from the evanescent region is suggestive of subsequent wave transmission. Transmitted waves
propagating from the critical layer can deposit their momentum, creating a region of EP-flux convergence (westward
acceleration). However, such over-reflection features become obscure from 3 January as the downward EPFz below the
evanescent region disappears. Moreover, the region of positive EPFD shifts to higher latitudes (60–90°N) than the region
where the upward-propagating WPW2 can reach (30–60°N). Therefore, the observed WPW2 amplification are not
satisfactorily explained through the over-reflection perspective.
Close inspection of the squared refractive index in Figure 5 also confirms that the wave resonance suggested by AB14
is less likely for the observed WPW2 explosion. Resonant wave events require a three-sided cavity of vertically
propagating PWs capable of trapping their energy. Such a cavity consists of two vertically oriented critical lines—one
in the midlatitudes and another in the polar regions—and a third horizontal one across the upper stratosphere. While
several localized regions of positive $n^2$ exist within the instability areas, obvious features indicative of wave cavity are
not identified. Furthermore, the characteristic EPFz behavior indicating wave resonance, that is, vertically instantaneous
EPFz (AB14), is not identified in Figure 2a.
Alternately, Song et al. (2020) demonstrated that the mesospheric EPW2 was generated by the zonally asymmetric
gravity wave (GW) forcing, namely the non-conservative source term ($Z'$) in the linearized perturbation QGPV equation.
We examine whether the rapid growth of the stratospheric WPW2 before the SSW21 onset is attributable to this
mechanism.
A linearized disturbance QGPV equation in log-pressure coordinates is as follows (Andrew et al., 1987):




$$\left(\frac{\partial}{\partial t} + \bar{u}\frac{\partial}{a\cos\phi\,\partial\lambda}\right)q' + v'\frac{\partial\bar{q}}{a\partial\phi} = \frac{1}{a\cos\phi}\left[\frac{\partial Y'}{\partial\lambda} - \frac{\partial(X'\cos\phi)}{\partial\phi}\right] + \frac{f_0}{\rho_0}\frac{\partial}{\partial z}\left[\rho_0\frac{Q'}{e^{\frac{\kappa}{H}z}\left(\frac{\partial\overline{T_0}}{\partial z} + \frac{\kappa\overline{T_0}}{H}\right)}\right], \quad (6)$$


$$q' \equiv \frac{1}{a^2\cos\phi}\left[\frac{1}{\cos\phi}\frac{\partial^2}{\partial\lambda^2} + \frac{\partial}{\partial\phi}\left(\cos\phi\frac{\partial}{\partial\phi}\right)\right]\psi' + \frac{1}{\rho_0}\frac{\partial}{\partial z}\left(\rho_0\frac{f_0^2}{N^2}\frac{\partial\psi'}{\partial z}\right), \quad (7)$$


$$\frac{\partial\bar{q}}{a\partial\phi} \equiv \frac{2\Omega\cos\phi}{a} - \frac{1}{a^2}\frac{\partial}{\partial\phi}\left[\frac{1}{\cos\phi}\frac{\partial(\bar{u}\cos\phi)}{\partial\phi}\right] - \frac{1}{\rho_0}\frac{\partial}{\partial z}\left(\rho_0\frac{f_0^2}{N^2}\frac{\partial\bar{u}}{\partial z}\right). \quad (8)$$



Here, $\lambda$ is the longitude, and $q'$ is the QGPV perturbation. $X'$ and $Y'$ denote the perturbation of the zonal and meridional
components of GW forcing from their zonal-mean, respectively. $Q'$ is the perturbation diabatic heating rate, and $\psi'$ is
the perturbation streamfunction ($\psi' = \phi'/f_0$, where $\phi'$ is the perturbation geopotential). The first bracketed term on the
right-hand side of Equation (6) is the nonconservative forcing term of the QGPV perturbation associated with the GW
drag (GWD). Therefore, we investigate whether the nonconservative GWD forcing defined by $Z'$ below is related to the
rapid enhancement of WPW2 by using the zonal and meridional components of the parameterized GWD data
(McFarlane 1987; Molod et al., 2015).


$$Z' = \frac{1}{a\cos\phi}\left[\frac{\partial Y'}{\partial\lambda} - \frac{\partial(X'\cos\phi)}{\partial\phi}\right] \quad (9)$$


Figure 6a presents the latitude-height cross sections of the zonally averaged $Z'$ magnitude ($|Z'|$) and the positive EPFD
of PW2 during the amplification period of WPW2. The upward propagating parameterized GWs are dissipated in regions
with strong vertical shears of the zonal-mean zonal winds (see Figure S1), yielding the zonally asymmetric GW forcings.
Accordingly, the zonal-mean $|Z'|$ is also identified above the strong shear region, where the positive EPFD is located.
However, due to the small magnitude of the GW forcing, $|Z'|$ above the positive EPFD region (1–5 hPa) is much smaller
than $|Z'|$ in the upper stratosphere and lower mesosphere (above 0.5 hPa), where $Z'$ became significant enough to
generate EPW2 in Song et al. (2020). More importantly, as evidenced from a series of polar stereographic plots of $Z'$
shown in Figure 6b, we cannot recognize an obvious ZWN2 structure. Therefore, we rule out the possibility of in situ
WPW2 generation driven by zonally asymmetric GW forcing as a nonconservative source of QGPV perturbation. Thus,
at least for the case of SSW21, our results support that BT/BC instability is the most likely source.

**3.4 Vortex Preconditioning: Double Westerly Jets**
The above findings lead us to examine the prewarming evolution of PNJ, which adjusts the vortex conducive to
instability. Figures 7a and 7b present the latitude-height cross sections of the zonal-mean zonal wind and the resolved
wave (RW) activities, respectively.
On 1–10 December 2020, the wind structure is similar to climatology, with a single maximum in the high-latitude
stratosphere. However, after the westerlies weaken over the following 10 days (11–20 December), the maximum moves
to the subtropical upper mesosphere (21–28 December). On 29 December, the wind structure largely deviates from the
climatology, consisting of two local maxima with comparable strength: one in the subtropical lower-mesosphere and the
other in the polar stratosphere. This so-called a double-jet configuration was also identified before the SSW09 onset
(Iida et al., 2014; RLO21). Along between the two maxima, the subtropical easterly progresses towards the polar
stratopause, which corresponds to a significant negative anomaly above the 95% confidence level. This abnormal



easterly completely separates the double-jets on 1 January, initiating shear instability (Figure 3b).
This is achieved through the critical-level interaction between the double westerly jets and RWs (Figure 7b). Around
the zero-wind line between the subtropical easterly and the polar westerly, RWs propagating from the mid-latitude
troposphere are critical-level filtered, exerting the statistically significant negative EPFD at the 99% confidence level.
This negative forcing migrates the subtropical easterly poleward, further separating the jets. Subsequent RWs cannot
propagate equatorward any further and are filtered within the poleward-shifted intervening region between the two jets,
depositing again the anomalously strong negative forcing. The polar stratopause easterlies attributed to this positive
feedback rapidly descend into 10 hPa and intensify dramatically beyond 80 m/s, causing exceptionally strong BT/BC
instability. The negative RW forcing is mostly attributed to PW1 (Figure S2), whereas RWs having ZWN greater than
1 contributed insignificantly or even counteracted (not shown).
In summary, vortex preconditioning for SSW21 is characterized by the double-jet configuration. By facilitating the
critical-level interaction with the tropospheric PW1, this wind structure migrates the subtropical stratospheric easterlies
into the polar stratopause, thereby initiating catastrophic vortex deceleration and adjusting the vortex toward explosive
unstable PW2 growth.

### 3.5 Destabilization of ZWN2 waves

While the westward-propagating nature of the unstable PW2 is explained in connection with the background easterlies,
it remains unclear why ZWN2 perturbations are predominantly amplified. One possibility is that the prevailing ZWN2
fluxes forced from the troposphere may have been instantaneously destabilized at all altitudes, dominating over other
waves. This speculation aligns with Hartmann's (1983) suggestion that predominant disturbances are more likely to be
enhanced than those of higher ZWNs, despite their larger growth rates. However, it is not the case because the localized
EPFz divergences in the stratosphere are decoupled from the troposphere (Figure 2a). Furthermore, the quasi-stationary
tropospheric PW2 are not allowed to enter the stratosphere across their critical layer, as evidenced by their convergence
near the zero-wind line (Figure 3a).
The more probable explanation is that WPW2 arise in situ within the destabilized stratosphere that nonlinearly interacts
with PW1. Hartmann (1983) found that with the presence of PW1, the barotropic instability of PNJ could enhance the
growth rates of shorter waves with similar phase speeds. Manney et al. (1991) identified similar destabilization of both
waves 2 and 3, but wave 2 in particular. Relevant features are identified in Figure 8, which presents Ertel's PV (EPV)
on the 1500 K isentropic surface (near 2 hPa). From 1 January, irreversible mixing associated with substantial PW1
dissipation (Figure 7b) causes vortex filamentation along the vortex edge, yielding two additional high EPV cores.
Concurrently, the initially localized negative EPV meridional gradient develops into a zonal-mean field, with the higher
(lower) EPV advected toward the lower latitudes (pole). With growing instability, the two localized high EPV cores
merge into one, exhibiting a ZWN2 pattern. Numerical experiments exploring the most unstable mode with respect to
the given zonal flow can provide further convincing evidence, but that is beyond the scope of this study.

### 4 Summary and Conclusion

During the SSW21 onset, an anomalous WPW2 growth appears, which eventually splits the polar vortex. Previous
studies have suggested that the enhanced ZWN2 fluxes originating from the tropospheric precursor events are
responsible for this stimulating PW2 activities. However, simultaneous enhancements in PW2 activities in the
tropopause and the mid-stratosphere are not explained solely by the vertical propagation of the tropospheric PW2. The
prominent westward-propagating PW2 in the mid stratosphere that differs from the quasi-stationary tropospheric PW2
complements this view.
This study demonstrates that the explosive WPW2 amplification occurs in situ within the polar stratosphere driven
toward BT/BC instability, where the easterlies rapidly descend from the stratopause including the critical layer of WPW2.
Vortex destabilization is induced as the abnormal double-jet structure having subtropical mesospheric and polar
stratospheric cores evolves toward SSW21 within just 7 days. Therefore, we suggest vortex preconditioning for SSW21
as the double-jet structure, which initiates vortex deceleration as well as tunes the vortex toward instability by facilitating
the critical-level interaction with the tropospheric PWs.



Our findings provide some key insights into preconditioning of SSWs. First, vortex destabilization is an inevitable
consequence of the zonal wind reversal to easterlies connected to the major SSWs. We found that all 26 major SSWs
for 42 years (selected following the CP07 definition) exhibit BT/BC instability associated with the prevalent easterlies
in the stratosphere at their onset (Figure S3). Given that an unstable flow supports the in situ PW explosion, which can
even shape the vortex geometry shortly before the SSW onset, we suggest to look in more detail into the influences of
BT/BC instability on the characteristics of SSW, including its onset, intensity, and duration. Second, the double-jets
structure is likely a stratospheric precursor that favors triggering SSW. Approximately 70% (19) of 26 major SSWs
exhibit this wind configuration within two weeks prior to their onset, despite variance in their occurrence timing (not
shown). The present case SSW21 that occurred under unfavorable tropical conditions (the westerly quasi-biennial
oscillation and weak convections) for SSW, reinforces this perspective. RLO21 also reported that this wind structure
and associated unstable PW generation are commonly identified in other SSW events. Therefore, the preceding double-
jets structure are worth examining in SSW studies to improve our understanding and predictability of SSWs. While this
study focuses on the evolution of the double-jet structure toward SSW, it would also be fruitful to investigate the
formation of such wind structure considering the interplay among PWs, GWs, and mean-flow (Iida et al., 2014; RLO21;
Sato and Nomoto, 2015).

**Data availability**

The MERRA2 data are available from the Global Modeling and Assimilation Office at NASA Goddard Space Flight
Center through the NASA GES DISC online archive (available online at https://doi.org/10.5067/WWQSXQ8IVFW8,
GMAO, 2015). All results made in this study can be provided by the corresponding authors upon request.

**Author contributions**

JHY, HYC, and MJK conceived the study. JHY conducted formal analysis and visualized the results. JHY wrote the
draft with a contribution from HYC and MJK.

**Competing interests**

The authors declare that they have no conflict of interest.

**Financial support**

This work is supported by a National Research Foundation of Korea grant funded by the South Korea government (20
21R1A2C100710212). The first author is supported by the Global PhD Fellowship Program (2019H1A2A1077307).




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







**Figure 1:** Time-height cross sections of (a) the zonal-mean zonal wind at 60°N (left) and polar cap temperature averaged over 60–90°N (right) and (b) the geopotential height (GPH) amplitude of the planetary waves (PWs) with zonal wavenumbers (ZWN) 1 (PW1, left) and 2 (PW2, right) at 60°N. The dark and bright pink (green) dots denote regions where the analyzed variable is algebraically smaller (larger) than its 42-year climatology by more than 1.96 and 2.57 standard deviations (STD), indicating that the variable is significantly anomalous at the 95 and 99% confidence levels, respectively. (c) Polar stereography series of the horizontal wind speed (shading) and GPH anomalies from their zonal-mean (contours) at 1 hPa (upper) and 10 hPa (lower) on 1–5 January. The red (blue) contour represents the positive (negative) value.

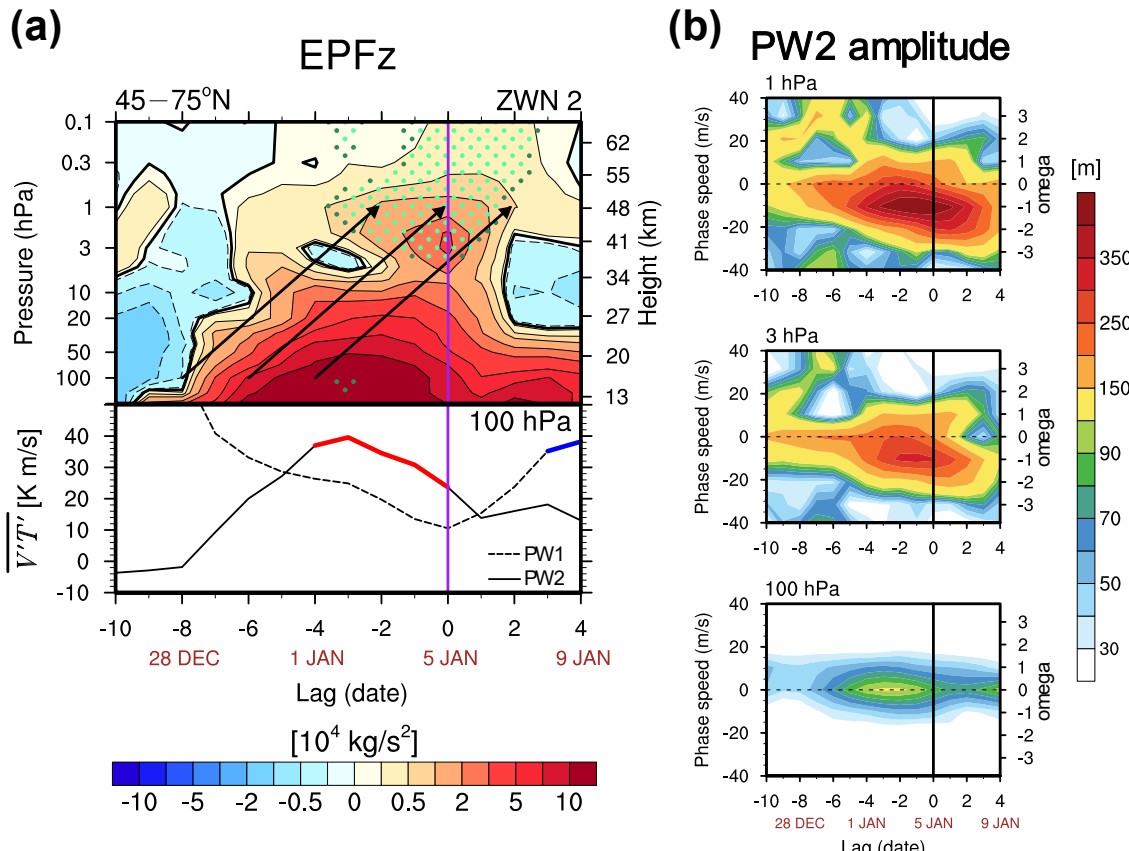

**Figure 2:** (a) Time-height cross sections of the vertical component of Eliassen-Palm fluxes (EPFz) of PW2 (upper) and time-series of eddy heat flux ($\overline{v'T'}$) of PW1 (dashed) and PW2 (solid) at 100 hPa (lower) averaged over 45–75°N. The overlaid blue (red) thick line denotes $\overline{v'T'}$ of PW1 (PW2) having a magnitude 1 STD greater than its climatology. (b) Time-zonal phase speed cross sections of the PW2 GHP amplitude at 1, 3, and 100 hPa averaged over 45–75°N. The purple and black vertical lines in (a) and (b), respectively, represent the onset date.







**Figure 3:** Latitude–height cross sections of (a) Eliassen-Palm fluxes (EP-fluxes, vectors) overlaid on their divergences (EPFD, colors) of PW2, (b) the meridional gradient of the quasi-geostrophic potential vorticity ($\bar{q}_y$, colors) overlaid by the positive EPFD of PW2 (red contour), (c) barotropic, and (d) baroclinic terms of Equation (3) in 1–5 January. The black contours present the zonal-mean zonal winds. The solid, dashed, and thick solid lines indicate positive, negative, and zero wind, respectively.







**Figure 4:** Latitude-height cross sections of (a–c) the three terms on the right-hand side of Equation (4) divided by $f^2$, (d) the inverse of the squared Brunt–Väisälä frequency $\frac{1}{N^2}$, (e) the vertical gradient of the zonal-mean zonal wind $\bar{u}_z$, (f), the vertical gradient of the squared Brunt–Väisälä frequency $N^2_z$, and (g) the vertical curvature of the zonal-mean zonal wind $\bar{u}_{zz}$ on 1–5 January 2021. The black contours present the zonal-mean zonal winds. The solid, dashed, and thick solid lines denote positive, negative, and zero wind, respectively.




**Figure 4:** (Continued).





**Figure 5:** Latitude-height cross sections of the negative $\bar{q}_y$ (mint shading) and positive refractive index squared ($n^2$, orange hatching) overlaid by PW2 EP-fluxes (vectors) and EPFD (contours, where the red and blue contours denote the positive and negative values, respectively) in 1–5 January 2021. The black contours present the zonal-mean zonal winds. The solid, dashed, and thick solid lines denote positive, negative, and zero wind, respectively.



**Figure 6:** (a) Latitude–height cross sections of the zonal-mean magnitude of nonconservative forcing ($Z'$, shading) overlaid by the positive EPFD of PW2 (red contour) during 1–5 January 2021. The black contours present the zonal-mean zonal winds where the solid, dashed, and thick solid lines denote positive, negative, and zero wind, respectively. (b) Polar stereography series of $Z'$ at 1 hPa altitude during the same period.






## (a) Zonal-mean U



Zonal-mean U [m/s]

## (b) EP-flux and EPFD

EPFD [m/s/day]

→ 750 m²/s²
EPFy, EPFz×333

**Figure 7:** Latitude-height cross sections of (a) the zonal-mean zonal winds averaged over 1–10, 11–20, 21–28 December 2020, and 29 December 2020–5 January 2021 (first row), daily from 29 December 2020 to 5 January 2021 (second to third row), and (b) EP fluxes (vectors) overlaid on the EPFD (colors) of the resolved waves. The black contours in (b) are the zonal-mean zonal winds. The contour specifications are the same as in Figure 3.



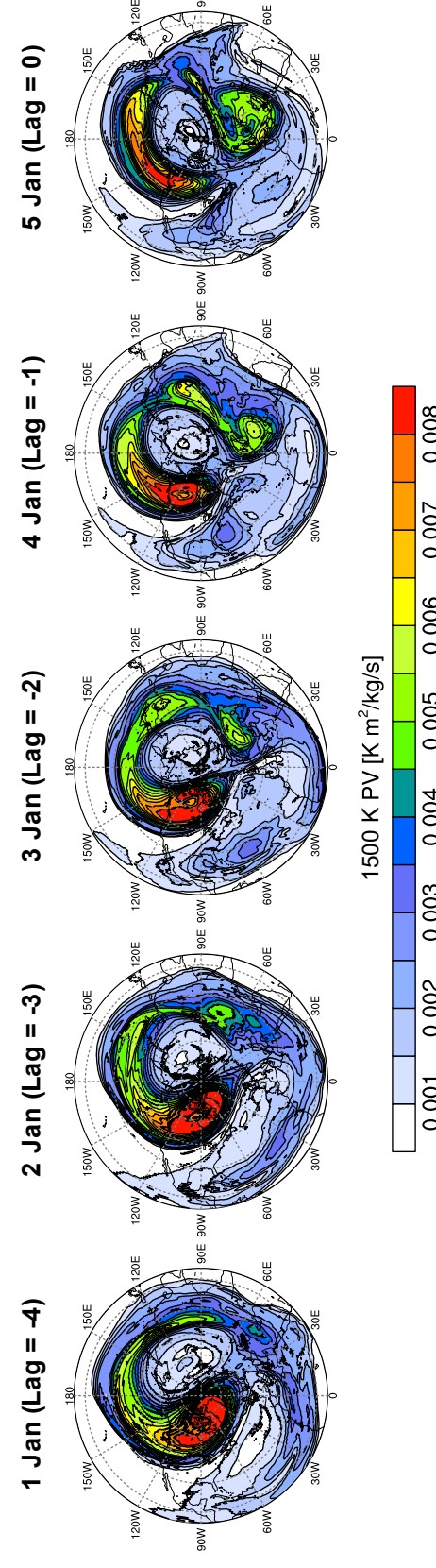

**Figure 8:** Time series of Ertel's potential vorticity at the 1500 K isentropic surface (~2 hPa).

460