# Peer review of "Vortex Preconditioning of the 2021 Sudden Stratospheric Warming: Barotropic/Baroclinic Instability Associated with the Double Westerly"

_EGUsphere, 2023_

## Author Response (AR1)

**Response to Reviewers' Comments**

**Ji-Hee Yoo, Hye-Yeong Chun, and Min-Jee Kang**

Department of Atmospheric Sciences, Yonsei University, Seoul, Korea

**August 8, 2023**

Dear Editor and Reviewers,

Thank you very much for your constructive comments/suggestions on this paper. We revised the manuscript to address all comments and suggestions and tried our best to improve the manuscript. Especially, to enhance readability, we have carefully chosen specific days which represent the key characteristics to be plotted in Figures 1c, 4, 5, and 6, instead of displaying data for every day throughout the entire period.

We answer questions of each Reviewer in the following paragraphs. Below, we indicate the original comment of the respective Reviewer in blue and our answer is denoted in black.

We would like for the current manuscript to be accepted to the ACP.

Sincerely,

Hye-Yeong Chun

**Response to Reviewer #1's Comments**

**General Comments:**

In this study, the authors focus on the vortex preconditioning of the 2021 SSW. The paper emphasized the impacts of the double jet cores on the 2021 SSW. I reviewed this paper when it was submitted to another journal, and most of my concerns in previous reviews have been well addressed. Therefore, I have no major comments, and only several very minor revisions are required.

Thank you very much for your careful review and valuable comments on this manuscript. We seriously considered your comments and posted responses to your comments/suggestions on the revised manuscript. We would like to clarify one thing that this manuscript has never been peer-reviewed before. When we submitted the previous version of the manuscript to Geophysical Research Letters, the editor of GRL suggested us for submitting it to a more general article journal rather than letters, without sending the manuscript to the reviewers.

**Minor Comments:**

1. L26: where => when

   It is modified as suggested [L26].

2. L29: The tense in the first half sentence and the second half sentence is not consistent. Suggest to use the simple present tense.

   It is modified as suggested from passive to active voice [L29].

3. L38: focuses => focuses on

   Because the verb "focuses" is used with "into" in this sentence, we decide to maintain the original sentence.

4. L228-242: This paragraph should be moved to the "method" section.

   Following the Reviewer's suggestion, the paragraph is moved to the "method" section in the revised manuscript [L94–108].

5. There are too many plots in one figure, and the readability is still not good. Figure 1c can be further condensed and only typical days are shown. Many plots in Figures 3–5 are also duplicated from day to day.

   Thank you for the valuable suggestion. In Figure 1c, we have chosen to display for 1, 3, and 5 January. Variables depicted in Figures 3 and 5 exhibit significant temporal variations, so we have decided to retain Figure 3 entirely and exclude only a plot for 1 January from Figure 5 [L220]. Conversely, the major feature that we would like to mention from Figures 4 and 6 remains consistent over time; hence, we only plot data for one day (3 January) as a representative example of the 5-day period [L189, L245].

**Response to Reviewer #2's Comments**

**General comment:**

This manuscript presents a study on the dynamical mechanisms behind the stratospheric vortex split during the stratospheric sudden warming (SSW) that took place on January 5th, 2021 using the reanalysis MERRA2. The authors present strong pieces of evidence that support the interpretation of baroclinic/barotropic instability in the upper stratosphere as the cause of the explosive growth of the planetary wave-2 that led to the vortex split. Alternative dynamical mechanisms are also explored, such as wave propagation from the troposphere, wave growth by asymmetric gravity wave forcing, or resonance, but none of them were found to be consistent with the flow and wave fluxes evolution during the SSW onset.

I find that the analysis is thorough and the figures are clearly interpreted, and that the results make a strong case for internal stratospheric dynamics as a source of explosive wave growth during SSWs. I have only a few very minor comments, I recommend publication.

Thank you very much for your careful review and valuable comments on this manuscript. We seriously considered your comments and posted responses to your comments/suggestions on the revised manuscript.

**Minor Comments:**

1. Line 47. This needs to be changed, the latest major warming happened last winter, in February 2023.

   Thank you very much for pointing out this! The expression "latest" is deleted [L47]

2. line 100: the level of 1 hPa is located around the stratopause, hence "upper stratosphere".

   Thank you for pointing out this. Following the Reviewer's comment, it is modified throughout the manuscript. [L12, 58, 115, 130, 134, 205, 306, 307].

3. Lines 127-133. It seems to me that there is a bit of over-interpretation of Fig. 2a, the time-height evolution of EPFz is not entirely inconsistent with the theoretical vertical group velocity. Given the evidence provided in Figs. 2b and 3 on the decoupling between the wave-2 in the lower and upper stratosphere, there is no need to stretch the argument in Fig. 2a. Besides, I find it more compelling the fact that there appears a local EPFz maxima above 5hPa and lags-1 to 1, which cannot be explained by linear upward propagation.

   We agree with the Reviewer's comment that *the time-height evolution of EPFz is not entirely inconsistent with the theoretical vertical group velocity,* as reflected in our original manuscript (Line 128–130), which states, *"The theoretical prediction of Rossby waves' vertical propagation well matches the vertical propagation of EPFz below 5 hPa, indicating that the bulk of ZWN2 fluxes propagate upward."* Although we appreciate your suggestion on Fig. 2a, we decided to keep the original statement, as we think that the

EPFz analysis shown in Fig. 2a presents another compelling piece of evidence in conjunction with the decoupling signatures between wave-2 in the lower and upper stratosphere (Figure 2b and 3).

Thank you very much for pointing out that a local EPFz maxima above 5 hPa between Lag = -1 to Lag =1 cannot be explained by linear upward propagation. While this feature was mentioned as *"the divergence continues with the locally maximized upward EPFz in 10–3 hPa"* in the original manuscript (Line 136–137), we reinforce this sentence to convey its significance following the Reviewer's comment in the revised manuscript (Line 150–153).